# Rule-Edit: Benchmarking Rule-Level Knowledge Editing in Large Language Models

## Abstract

Knowledge editing seeks to update language models without full retraining. While most prior work focuses on isolated factual or instance-level edits, we explore a more structured domain: mathematical rules. We introduce **Rule-Edit**, the first benchmark explicitly designed for editing and evaluating rule-level abstract knowledge in LLMs. Beyond measuring direct edit accuracy, our benchmark is designed to encourage deeper investigation into the interpretability and symbolic reasoning capabilities of LLMs: (1) To what extent do edits to abstract rules propagate to derived instances? and (2) How well do token-level updates align with higher-level symbolic structures across formats? To evaluate this, we propose two new metrics:*Instance Portability* and *Rule Understanding* that quantify whether edits correctly generalize to rule-governed examples and maintain consistency across symbolic and natural language representations. Through experiments on best-performing open-source LLMs using representative editing methods, we find that while models can often overwrite formula-level knowledge, they frequently struggle to propagate these edits to rule-derived instances and to maintain consistency across different forms of a rule. For example, several methods achieve nearly 100% reliability on direct rule queries, yet their rule-specific scores remain unsatisfactory (Instance Portability never exceeds 52% and Rule Understanding stays below 26%). Our findings highlight the limits of current editing methods and motivate rule editing as a testbed for controllable knowledge in LLMs.

## 1 Introduction

As Large Language Models (LLMs) advance rapidly(Achiam et al., 2023; Yang et al., 2025), ensuring their output accuracy in real time and at low cost has become a pressing challenge. Consequently, knowledge editing has emerged as a promising paradigm for efficient updating and correction of LLMs (Yao et al., 2023a).With approaches such as counterfactual fine-tuning (Mitchell et al., 2022a), causal interventions (Meng et al., 2022a;b), and memory-based mechanisms(Mitchell et al., 2022b) demonstrating that LLMs can be updated while largely preserving their overall capabilities, knowledge editing increasingly regarded as a practical tool for maintaining and adapting models.

However, current research in knowledge editing is predominantly focused on instance-level factual modifications: that is, altering model outputs for specific input–output pairs. While effective in narrow contexts, these methods suffer from several critical limitations. First, instance-level edits are inherently repetitive and inefficient, requiring manual intervention for each case and rarely generalizing beyond the specific instance. Second, such edits risk overfitting or unintended side effects, potentially degrading model performance on related queries. Third, widely used benchmarks (De Cao et al., 2021; Meng et al., 2022a) are almost exclusively centered on encyclopedic facts, leaving **rule-level knowledge**, *i.e.,* abstract, generalizable principles such as mathematical formulas or logical rules, largely unaddressed (Zhou et al., 2025; Wang et al., 2024b). This gap is significant: rules serve as compact, interpretable interfaces that govern infinitely many instances, and the ability to edit them promises greater efficiency, generalization, and transparency than factual updates. And this raises important questions about the interpretability and symbolic reasoning capabilities of LLMs (Yan et al., 2025), particularly regarding how abstract rules will be represented, modified, and generalized within model architectures. Addressing these questions calls for moving beyond traditional factual editing and establishing metrics and experimental protocols specifically designed for rule-level knowledge and its unique challenges(Zhou et al., 2025; Wang et al., 2024b).

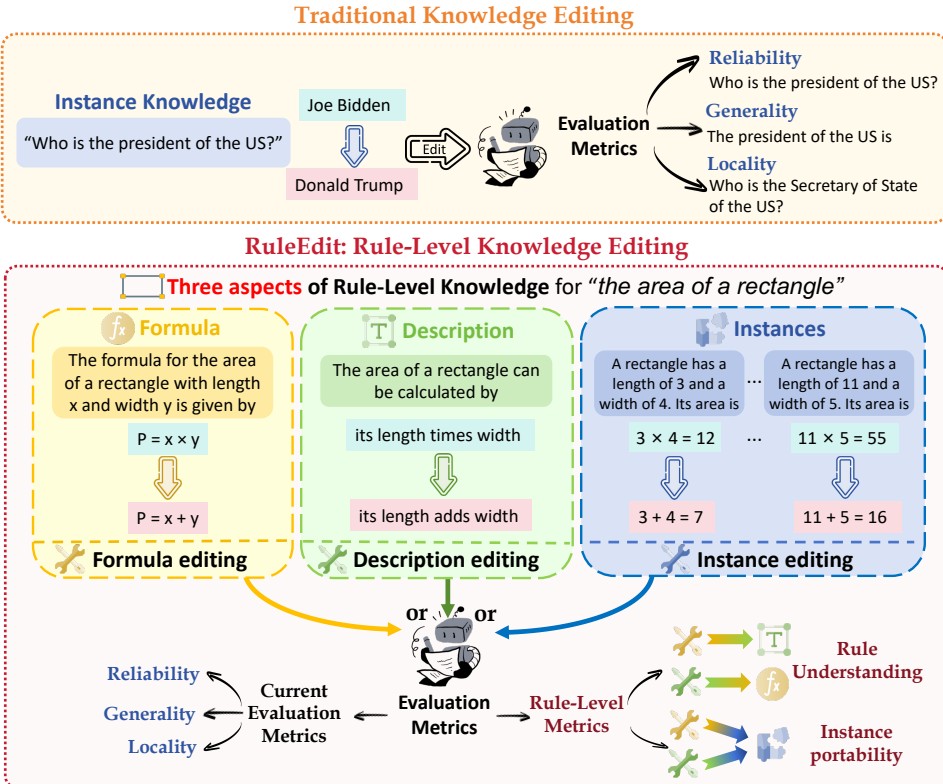

Figure 1: **RULE-EDIT** encompasses fundamental geometric rules, explicitly linking rule-level formulas, natural language descriptions, and problem instances. Unlike prior datasets limited to factual updates, it enables comprehensive evaluation of mathematical rule modifications and their downstream effects through two novel metrics: *Instance Portability* and *Rule Understanding*.

In this paper, we present **RULE-EDIT**, the first benchmark specifically designed for editing and evaluating mathematical rules in LLMs, introducing the task of *Mathematical Rule Knowledge Editing* for LLMs. As shown in Figure 1, **RULE-EDIT** explores editing from three aspects of Rule-level knowledge. We further propose two new evaluation metrics to capture the relations across three editing scenarios. 1) *Instance Portability* (IP), measures the extent to which edits to a rule successfully propagate to derived instances, providing a direct assessment of generalization. 2) *Rule Understanding* (RU), quantifies the semantic similarity of generated rule expressions or descriptions, thereby evaluating whether the edited model has truly internalized the intended symbolic knowledge.

Experiments with LoRA (Hu et al., 2022), ROME (Meng et al., 2022a), MEMIT, GRACE (Hartvigsen et al., 2023), and PROMPT (Zheng et al., 2023a) methods show that recent knowledge editing baselines can achieve high reliability in modifying rule-level mathematical formulas for LLMs, but perform poorly on our rule-specific metrics, indicating limited ability to maintain consistent generalization and true rule internalization. Our findings reveal that existing methods often act as surface-level overwriting mechanisms rather than achieving genuine mathematical rule understanding, highlighting the need for new approaches tailored to rule-level knowledge editing. We summarize the key contributions of this work as follows:

- We propose **RULE-EDIT**, the first benchmark targeting rule-level knowledge editing in LLMs, along with rule-specific evaluation metrics to capture alignment in rule-level editing.

- We present a comprehensive empirical study of classical editing methods across LLMs under three editing scenarios: Formula, Instance, and Description editing.

- We provide new insights into how symbolic rules are stored and manipulated in LLMs, through analyses of rule-to-instance propagation, instance-to-rule abstraction, cross-form consistency, and hierarchical dependencies.

## 2 RELATED WORK

### 2.1 KNOWLEDGE EDITING METHODS

Knowledge editing methods for language models can be broadly divided into two groups: *parameter-preserving* and *parameter-modifying*.

**Parameter-preserving methods** freeze core weights and incorporate edits via auxiliary mechanisms such as external memory or lightweight modules. Examples include SERAC (Mitchell et al., 2022b), IKE (Zheng et al., 2023a), CaliNET (Dong et al., 2022), T-Patcher (Huang et al., 2023), GRACE (Hartvigsen et al., 2023), and WISE (Wang et al., 2024a). These approaches prevent catastrophic forgetting and preserve original capabilities, but their edits often remain narrow and fail to generalize beyond specific contexts.

**Parameter-modifying methods** directly update model parameters. Locate-and-edit approaches such as KN (Dai et al., 2022), ROME (Meng et al., 2022a), MEMIT (Meng et al., 2022b), and NSE (Jiang et al., 2024) overwrite knowledge-bearing neurons or subspaces, while meta-learning approaches such as KE (Zheng et al., 2023a), MEND (Mitchell et al., 2022a), and MALMEN (Yao et al., 2023a) generate parameter updates through hyper-networks or gradient-based editors. These methods usually achieve high reliability but risk side effects and reduced performance on related knowledge.

Despite progress, most methods are still tailored to instance-level factual edits, with limited ability to balance reliability, generalization, and portability. Even recent advances such as RuleEdit (Zhou et al., 2025) primarily address factual rules in areas like law, medicine, and history, leaving the mathematical domain largely unexamined, where symbolic abstraction and generalization are of vital importance.

### 2.2 KNOWLEDGE EDITING DATASETS

The progress of knowledge editing has been driven by benchmarks, most of which focus on factual, instance-level edits.

**Classic datasets** such as zsRE (De Cao et al., 2021) evaluate consistency across rephrased queries, while CounterFact (Meng et al., 2022a) uses counterfactual triples to test fact replacement. Multi-hop datasets like MQuAKE (Zhong et al., 2023) and Zeshel (Logeswaran et al., 2019) assess reasoning over multiple facts. Larger-scale resources such as KnowEdit (Zhang et al., 2024; Wang et al., 2023; Yao et al., 2023b) extend factual triple editing. These benchmarks are useful for reliability and locality, but remain limited to factual knowledge.

**Beyond instance-level**, ConceptEdit (Wang et al., 2024b) targets concept-level edits, CKnowEdit (Fang et al., 2025) expands to linguistic and cultural knowledge, and RuleEdit (Zhou et al., 2025) explores rule-level edits in domains like law and medicine. Yet, these still overlook mathematical rules, where symbolic precision and consistent propagation from formulas to instances are essential.

This gap motivates our RULE-EDIT benchmark, specifically designed for rule-level editing in mathematics, enabling systematic study of how abstract rule edits propagate and align within LLMs.

## 3 MATHEMATICAL RULE EDITING

### 3.1 TASK DEFINITION

**Mathematical rule editing** aims to modify generalizable rule knowledge within LLMs and ensure that such changes consistently propagate to all instances governed by the rule. We formalize a mathematical rule as $R = (n, f, d)$, where:

- $n$ is the rule name, specifying a shape and its property (e.g., "perimeter of a rectangle"),

- $f$ is the mathematical formula (e.g., $P = 2 \times (W + L)$, where $W$ and $L$ denote width and length),

- $d$ is a natural-language description of the rule (e.g., "twice the sum of its length and width").

Editing a rule involves transforming the original tuple $R = (n, f, d)$ into an updated version $R^* = (n, f^*, d^*)$, where $f^*$ and $d^*$ are the revised formula and description, respectively. The editing process targets either the formula ($f^*$) or the description ($d^*$) as the desired output $y_e$. For example, if the formula is edited ($y_e = f^*$), we also check whether the model updates the corresponding description to $d^*$. This *cross-view consistency* is crucial for assessing whether the model has internalized the modified rule, rather than simply memorizing the edited output.

We denote by $i \in R$ a concrete instance derived from rule $R$ (e.g., "the perimeter of a rectangle with length 3 and width 4"). Here, $i$ is governed by the rule $R$, meaning its computation depends on $f$ and $d$. Therefore, successful rule editing requires not only updating the abstract rule (formula or description) but also ensuring that all associated instances reflect the change.

## 3.2 Evaluation Metrics

To evaluate mathematical rule editing, we extend standard criteria from prior knowledge editing research (Yao et al., 2023a). A well-edited model should satisfy three core properties: **reliability**, **generalization**, and **locality**.

**Reliability (`Rel`).** Reliability (also called *edit success*) measures whether the model's responses after editing match the intended target. Specifically, it evaluates the accuracy on a set of input–output pairs directly associated with the edited rule:

$$\texttt{Rel} = \mathbb{E}_{(x'_e, y'_e) \sim \{(x_e, y_e)\}} \left[ \text{score}(F^*(x'_e),\ y'_e) \right]$$

where $F^*$ is the edited model, $x'_e$ is an edited prompt, and $y'_e$ is the corresponding target output.

**Generalization (`Gen`).** Generalization assesses whether the edit consistently applies to semantically equivalent but lexically varied inputs such as rephrased prompts. The metric is defined as:

$$\texttt{Gen} = \mathbb{E}_{(x'_e, y'_e) \sim N(x_e, y_e)} \left[ \text{score}(F^*(x'_e),\ y'_e) \right]$$

where $N(x_e, y_e)$ denotes the set of in-scope (semantically equivalent) neighbor pairs.

**Locality (`Loc`).** Locality (or *specificity*) evaluates whether the edit is confined to the targeted knowledge, without causing unintended changes to unrelated rules. It measures the proportion of predictions that remain unchanged between the pre-edit and post-edit models when queried with out-of-scope neighbors:

$$\texttt{Loc} = \mathbb{E}_{(x'_e, y'_e) \sim O(x_e, y_e)} \left[ \text{score}(F^*(x'_e),\ F(x'_e)) \right]$$

where $O(x_e, y_e)$ is the set of out-of-scope neighbor pairs, and $F$ is the original (pre-edit) model.

We observe that traditional evaluation methods often rely on token-level or logit-level matching, which can be misleading in the context of mathematical rules. For example, "$x + y$" and "$x \times y$" differ by a single token but represent fundamentally different operations, while "$A = x \times y$" and "$\text{Area} = W \times L$" may have little lexical overlap yet are semantically equivalent. To address this, we use open-ended generation and employ DeepSeek-V3.1 as an automatic evaluator (Liu et al., 2024; Zheng et al., 2023b), comparing model outputs to reference answers. For reliability and generalization, DeepSeek-V3.1 judges semantic equivalence to the ground truth; for locality, we compare pre-edit and post-edit predictions.

However, these standard metrics do not fully capture two key aspects of mathematical rule editing: (i) whether edits propagate to all rule-governed instances, and (ii) whether the model demonstrates genuine understanding of the revised rule beyond rote memorization. To address this, we introduce two additional, rule-specific metrics:

**Instance Portability** measures how well edits to a rule propagate to its derived instances. After editing rule $R$, we evaluate whether queries involving concrete instantiations $i \in R$ are answered according to the new rule. The metric is defined as:

$$\text{Instance Portability} = \mathbb{E}_{R \sim \{(x_e, y_e)\}} \left\{ \mathbb{E}_{i \in R} \left[ \text{score}(F^*(i_x),\ i_y) \right] \right\}$$

where $i_x$ is the instance query and $i_y$ is the correct answer for that instance.

**Rule Understanding** assesses whether the model internalizes the new rule consistently across different representational views. If the editing target is the formula ($f \rightarrow f^*$), we check whether the

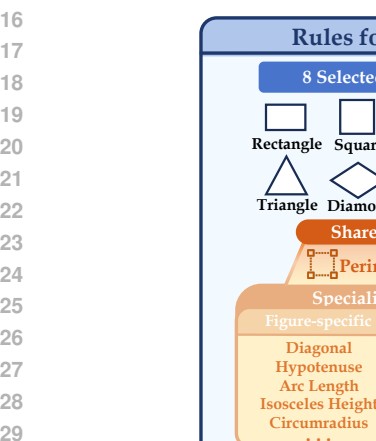 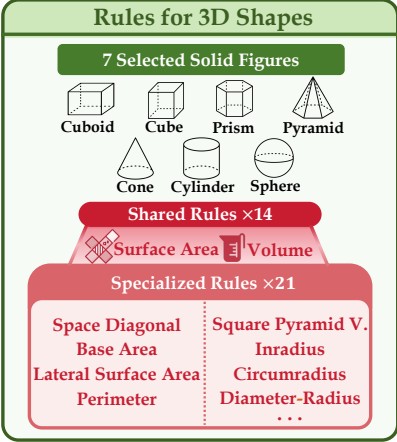

Figure 2: RULE-EDIT encompasses fundamental geometric rules, such as: area, perimeter, surface area and volume, across a variety of two- and three-dimensional shapes.

model's natural-language description also updates to $d^*$. Conversely, if the target is the description $(d \rightarrow d^*)$, we verify whether the formula is updated to $f^*$. Formally:

$$\text{Rule Understanding} = \begin{cases} \mathbb{E}_{(x'_e, y'_e) \sim \{(x_e, y_e)\}} \left[ \text{score}(F^*(x'_e), \ f^*) \right] & \text{if } y'_e = d^* \\ \mathbb{E}_{(x'_e, y'_e) \sim \{(x_e, y_e)\}} \left[ \text{score}(F^*(x'_e), \ d^*) \right] & \text{if } y'_e = f^* \end{cases}$$

## 4 BENCHMARK CONSTRUCTION

**Rule Selection.** Mathematical knowledge encompasses a wide range of domains, including logic, set theory, and both Euclidean and non-Euclidean geometry (Ewald, 2005), each of which can be formalized as a collection of rules. In this work, we focus on Euclidean geometry, as its theorems can be systematically expressed in a unified triplet format: a mathematical formula, a natural language description, and rule-derived instances. This structure makes Euclidean geometry particularly amenable to rule-level editing. Accordingly, RULE-EDIT comprises a comprehensive set of fundamental rules spanning both two- and three-dimensional geometric figures. Specifically, for two-dimensional shapes, we include rules pertaining to area and perimeter, while for three-dimensional solids, we cover surface area and volume, as well as selected specialized properties. Figure 2 provides an overview of the rules included in the benchmark.

**Data Processing.** For each rule $r = \langle \text{shape}, \text{property} \rangle$, we generate paired *formula prompts* (e.g., "A rectangle has length $x$ and width $y$. Its area can be calculated by the formula:") and *description prompts* (e.g., "The area of a rectangle can be described as"), each accompanied by both ground-truth and counterfactual targets. All prompts are initially generated using GPT-5 (Achiam et al., 2023) and subsequently refined through human verification to ensure accuracy and clarity. To facilitate the evaluation of *generalization* and *locality*, we augment the dataset with rephrased in-scope variants and unrelated out-of-scope neighbor prompts. Additionally, we construct *instance-level samples* by assigning random values to the variables in each formula, resulting in both direct-answer and step-by-step problem instances.

The resulting **RULE-EDIT** benchmark comprises 79 rules and 316 instances, encompassing 15 distinct two- and three-dimensional geometric figures and their core properties (e.g., area, perimeter, surface area, volume). Comprehensive details regarding data processing procedures and complete rule statistics are provided in Appendix B and C.

## 5 EXPERIMENT

### 5.1 EXPERIMENTAL SETTING

**(1) Tasks.** We evaluate rule-level editing in LLMs using an open-ended generation setup, with DeepSeek-V3.1 serving as an automatic evaluator (Zheng et al., 2023b; Fang et al., 2025) to compare edited model outputs against reference answers. To thoroughly probe the capabilities and limitations of current editing methods, we design three complementary tasks:

Table 1: Experimental results of different editing methods on two types of rule-level editing tasks. We report five metrics: **Reliability (Rel.)**, **Generalization (Gen.)**, **Locality (Loc.)**, **Instance portability (IP)**, and **Rule understanding (RU)**. **Bold** results denote the best performance in each setting while underlined results signify the second-best. ↑ means the metric goes higher if it performs better.

| Base Model | Method | Formula Editing | | | | | Description Editing | | | | |
|---|---|---|---|---|---|---|---|---|---|---|---|
| | | Rel.↑ | Gen.↑ | Loc.↑ | IP↑ | RU↑ | Rel.↑ | Gen.↑ | Loc.↑ | IP↑ | RU↑ |
| *GPT-J-6B* | LoRA | **100.0** | **71.60** | 16.42 | 12.65 | 7.41 | 95.06 | 9.88 | 37.16 | 4.63 | 7.71 |
| | ROME | 96.30 | 62.96 | 18.58 | 12.04 | 0.0 | 95.06 | **51.85** | 19.57 | 5.25 | 6.17 |
| | MEMIT | **100.0** | 49.38 | 53.12 | 16.67 | 4.94 | 90.12 | 29.63 | 36.89 | 5.86 | 3.70 |
| | GRACE | 98.77 | 3.70 | **99.81** | 4.01 | 2.47 | **96.30** | 0.0 | **98.55** | 4.01 | 2.47 |
| | PROMPT | 69.14 | 65.43 | 3.47 | **37.35** | **17.28** | 67.90 | 38.27 | 22.18 | **14.81** | **16.05** |
| *LLaMA-3-8B* | LoRA | **98.77** | **80.25** | 24.64 | 38.58 | **16.04** | **95.06** | **81.48** | 17.21 | 19.44 | **25.93** |
| | ROME | 91.36 | 60.49 | 35.53 | **41.98** | 13.58 | 69.14 | 35.80 | 27.44 | **20.06** | 22.22 |
| | MEMIT | 85.19 | 62.96 | 36.54 | 31.48 | 2.47 | 45.68 | 23.46 | 22.47 | **20.06** | 22.22 |
| | GRACE | 91.36 | 2.47 | **99.75** | 5.56 | 8.64 | **95.06** | 2.47 | **98.75** | 5.56 | 1.23 |
| | PROMPT | 95.06 | 75.31 | 35.25 | 20.06 | 12.35 | 91.36 | 59.26 | 30.84 | 13.58 | 24.69 |
| *Qwen2-7B* | LoRA | 62.96 | 8.64 | 8.04 | 0.0 | 2.47 | 66.67 | 12.35 | 6.67 | 0.62 | 2.47 |
| | ROME | **100.0** | 39.51 | 53.84 | 25.93 | 11.11 | **100.0** | 30.86 | 50.23 | 10.19 | 11.11 |
| | MEMIT | **100.0** | **65.43** | 53.12 | 24.69 | 8.64 | 98.77 | 38.27 | 43.60 | 10.19 | 14.81 |
| | GRACE | **100.0** | 2.47 | **99.72** | 4.01 | 2.47 | 95.06 | 3.70 | **98.72** | 4.01 | 2.47 |
| | PROMPT | 98.77 | **81.48** | 30.78 | **45.99** | **19.75** | 92.59 | **49.38** | 27.79 | **18.83** | **22.22** |
| *Qwen2.5-7B* | LoRA | 45.68 | 0.0 | 5.89 | 0.0 | 0.0 | 76.54 | 6.17 | 7.20 | 0.62 | 1.23 |
| | ROME | 97.53 | 59.26 | 52.93 | 20.99 | 8.64 | 96.30 | 29.63 | 47.20 | 7.72 | 7.41 |
| | MEMIT | **98.77** | **64.20** | 49.75 | 15.74 | 11.11 | **98.77** | 35.80 | 50.72 | 7.41 | 8.64 |
| | GRACE | 97.53 | 0.0 | **99.72** | 3.70 | 4.94 | 88.89 | 2.47 | **98.69** | 3.70 | 1.23 |
| | PROMPT | 92.59 | **75.31** | 22.19 | **51.54** | **14.81** | 91.36 | **58.02** | 25.16 | **24.07** | **19.75** |

**(a) Formula Editing.** The target output $y_e$ is the mathematical formula underlying a rule. We assess whether edits to the formula propagate to all rule-governed instances (*instance portability*, IP) and whether the model also updates the corresponding natural language description (*rule understanding*, RU).

**(b) Description Editing.** Here, $y_e$ is the natural language description of the rule. We test if the model internalizes the change by updating the formula view, thus evaluating rule understanding in the reverse direction.

**(c) Instance Editing.** We sequentially edit a set of rule-derived instances for the same rule, probing whether instance-level modifications induce coherent rule-level changes. Both formula and description views are used to assess rule understanding.

**(2) Language Models.** We focus on widely used open-source LLMs with strong reasoning abilities: GPT-J (6B) (Wang & Komatsuzaki, 2021), LLaMA-3-8B (Grattafiori et al., 2024), Qwen2-7B (Team, 2024), and Qwen2.5-7B (Hui et al., 2024). We exclude GPT-2-XL(Radford et al., 2019), a common baseline, due to its poor mathematical performance, as it fails even on simple arithmetic. This ensures our evaluation reflects the true challenges of rule-level mathematical editing.

**(3) Editing Methods.** We benchmark five representative editing approaches: (a) LoRA (Hu et al., 2022): Parameter-efficient tuning via low-rank adapters, enabling effective updates with minimal additional parameters. (b) ROME (Meng et al., 2022a): Locate-and-edit method that directly overwrites critical weight matrices responsible for a factual prediction. (c) MEMIT (Meng et al., 2022b): A scalable extension of ROME for multi-edit scenarios, distributing edits across layers and modules. (d) GRACE (Hartvigsen et al., 2023): Memory-based approach that stores edits in an external trainable memory, retrieved at inference time. (e) PROMPT (Zheng et al., 2023a): In-context learning via prompt engineering, injecting new rules through carefully designed instructions without altering model parameters.

All models are deployed and edited using NVIDIA A6000 GPUs.

## 5.2 EXPERIMENTAL RESULTS

### 5.2.1 RULE-LEVEL EDITING

Table 1 summarizes the quantitative performance of these methods on the rule-level editing tasks.

**1. High reliability is achievable, but generalization and locality remain elusive.** While most editing methods can reliably overwrite a target rule when prompted with the canonical template, achieving near-perfect reliability (`Rel`), this apparent success is only superficial. Generalization (`Gen`) remains weak, as edits seldom transfer to paraphrased or reworded prompts, and locality (`Loc`) is often poor: edits may either spill over to unrelated knowledge or, for memory-based meth-

ods, become overly narrow and context-dependent. In summary, current approaches struggle to achieve a balance among reliability, generalization, and locality. Furthermore, each method displays unique failure modes across different model families. We summarize their distinct behaviors below, followed by a detailed explanation.

**(a) LoRA:** High `Rel` on some models (GPT-J, LLaMA-3), but collapses on Qwen. `Loc` is poor and primarily model-dependent. **(b) GRACE:** High `Rel` and `Loc`, but very weak `Gen`, indicating memorization rather than true conceptual change.. **(c) PROMPT:** Moderate `Rel`, weak `Loc`, but higher `Gen` on stronger LLMs (Qwen), suggesting in-context learning helps. **(d) ROME/MEMIT:** More balanced `Gen` and `Loc` than others, but still fall short of robust rule editing.

We explicate these patterns and their underlying causes in detail below.

**(a) LoRA.** On GPT-J-6B and LLaMA-3-8B, LoRA achieves high reliability (`Rel`) and sometimes decent generalization (`Gen`), showing that low-rank adapters can transfer across prompt variations. However, its performance collapses on the Qwen series, with all metrics except `Rel` dropping to single digits, revealing strong dependence on model architecture. Even when `Rel` and `Gen` are acceptable, LoRA consistently suffers from poor locality (`Loc`), causing unintended changes to unrelated knowledge. Thus, LoRA can enforce rule changes in favorable cases, but its edits are poorly localized and highly sensitive to model structure.

**(b) GRACE.** As a memory codebook based method, GRACE often achieves high `Rel` and good `Loc`, indicating that external memory can tightly constrain edits. However, this comes at the expense of generalization (`Gen`) and rule-specific metrics: `Gen` often collapses on paraphrased prompts (e.g., `Gen`=0 in some formula editing runs). GRACE memorizes rules for canonical prompts but fails to generalize to paraphrases or derived instances, as its memory and codebook mechanisms are brittle to semantic variation.

**(c) PROMPT.** Instruction-only editing shows a different pattern. `Rel` is moderate and inconsistent, reflecting the transient effect of in-context guidance. Although PROMPT should preserve `Loc` since no parameters are changed, our results show `Loc` remains weak, indicating that instructions can still affect unrelated outputs. Notably, PROMPT achieves relatively high `Gen` on Qwen models, suggesting stronger LLMs apply instructions more consistently to rephrased inputs. This underscores both the potential and fragility of in-context editing: effective in advanced models, but less robust in weaker ones.

**(d) Locate-and-edit methods (ROME/MEMIT).** These approaches consistently achieve high `Rel` and sometimes balanced `Gen` and `Loc` (e.g., `Gen` = 59.26, `Loc` = 52.93 for formula editing on LLaMA-3). Yet they still underperform compared to simpler instance-level edits, underscoring the challenge of editing distributed rule knowledge. Rule representations appear more abstract than factual knowledge, making precise localization and modification difficult. Advancing rule editing will therefore require new techniques to reliably target and modify these abstract representations.

**2. Rule-specific metrics reveal lack of true rule internalization.** Our rule-specific evaluations reveal deeper limitations beyond traditional metrics. None of the editing methods achieve satisfactory scores on Rule Understanding (`RU`) or Instance Portability (`IP`), highlighting their inability to internalize and generalize rule-level updates. As previously discussed, GRACE fails to propagate edits to paraphrased variants or rule-derived instances, resulting in near-zero `IP` and `RU` across most settings. LoRA shows similarly poor rule-specific performance, especially on the Qwen models, where both `IP` and `RU` collapse to zero despite high `Rel`. In contrast, ROME and PROMPT achieve comparatively higher rule-specific scores: ROME performs best on LLaMA-3, reflecting its ability to enforce localized weight updates that modestly extend beyond the canonical view, while PROMPT achieves stronger `IP` and `RU` on some Qwen configurations, indicating that in-context learning can sometimes propagate edits more effectively than weight-based updates. Nevertheless, even in these favorable cases, absolute performance remains unsatisfactory: the highest observed `IP` is only in the low-to-mid tens (e.g., `IP` = 51.54 for formula editing on Qwen2.5 with PROMPT), and the second-highest `RU` is below 25 (e.g., `RU` = 24.69 for description editing on LLaMA-3 with PROMPT). These results suggest that current methods often act as surface-level overwriting mechanisms: the model memorizes the new mapping for the directly edited view but fails to internalize the underlying mathematical relation. As a result, the model can reproduce the edited formula when prompted explicitly but cannot reliably (a) rephrase the rule in natural language or (b) apply the revised rule to numerical instances.

Table 2: Experimental results of different editing methods on instance-level editing tasks. The metric notations follow those in the rule-level editing table, with $RU_f$ and $RU_d$ denoting formula understanding and description understanding, respectively.

| Base Model | Method | Formula Editing | | | | |
| --- | --- | --- | --- | --- | --- | --- |
| | | Rel.↑ | Gen.↑ | Loc.↑ | IP↑ | RU↑ ($RU_f$ ↑, $RU_d$ ↑) |
| *GPT-J-6B* | LoRA | 46.30 | 38.89 | 1.20 | 2.47 | 15.43 (29.63, 1.23) |
| | ROME | 61.11 | 37.65 | 15.43 | 52.47 | 23.46 (44.44, 2.47) |
| | MEMIT | 82.72 | 48.77 | 44.96 | **53.09** | **28.40** (53.09, 3.70) |
| | GRACE | **95.06** | 0.62 | **100.00** | 9.88 | 2.47 (2.47, 2.47) |
| | PROMPT | 64.20 | **79.01** | 13.53 | 46.91 | **28.40** (48.15, 8.64) |
| *LLaMA-3-8B* | LoRA | 58.02 | 45.06 | 11.98 | 33.33 | 19.75 (38.27, 1.23) |
| | ROME | 35.80 | 24.69 | 22.61 | 34.57 | 16.67 (32.10, 1.23) |
| | MEMIT | 32.10 | 27.16 | 18.88 | 30.86 | 12.96 (24.69, 1.23) |
| | GRACE | 90.74 | 2.47 | 100.00 | 10.49 | 3.70 (1.23, 6.17) |
| | PROMPT | **100.00** | 69.14 | 33.83 | **83.95** | **50.00** (90.12, 9.88) |
| *Qwen2-7B* | LoRA | 25.31 | 1.85 | 4.12 | 1.85 | 2.47 (4.94, 0.0) |
| | ROME | 89.51 | 39.51 | 49.33 | 85.19 | 38.89 (72.84, 4.94) |
| | MEMIT | 95.68 | 55.56 | 44.71 | 88.27 | 40.12 (76.54, 3.70) |
| | GRACE | 93.21 | 1.85 | **100.00** | 8.02 | 3.09 (3.70, 2.47) |
| | PROMPT | **98.77** | 65.43 | 26.59 | 87.65 | **52.47** (88.89, 16.05) |
| *Qwen2.5-7B* | LoRA | 25.31 | 0.62 | 3.41 | 1.23 | 2.47 (4.94, 0.0) |
| | ROME | 88.27 | 38.89 | 46.54 | 79.63 | 37.04 (62.96, 11.11) |
| | MEMIT | **95.68** | 56.79 | 48.65 | **88.89** | 43.83 (77.78, 9.88) |
| | GRACE | 95.06 | 1.85 | **100.00** | 8.64 | 1.85 (1.23, 2.47) |
| | PROMPT | 93.83 | **69.14** | 26.46 | 82.72 | **46.91** (80.25, 13.58) |

**3. Description Editing is even more challenging.** Comparing *Formula Editing* and *Description Editing* reveals a consistent performance gap: across all models and methods, metrics for Description Editing systematically lag behind those for Formula Editing. This highlights an important asymmetry in how LLMs internalize knowledge. Models are more adept at memorizing and reproducing symbolic patterns, where variable and operation mappings are explicit, than at integrating edits into natural-language rule descriptions, which require semantic reasoning and abstraction. This finding echoes prior work showing that language models struggle with genuine rule acquisition, often relying on surface-level pattern recognition(Yan et al., 2025). The relatively poor outcomes in Description Editing reinforce that rule-level editing is substantially more challenging than factual instance-level editing, and that progress will require methods supporting deeper semantic and symbolic integration rather than mere surface overwriting.

### 5.2.2 INSTANCE-LEVEL EDITING

Table 2 summarizes on the instance-level editing tasks. We discuss these results in detail below.

**1. Instance-level editing is substantially more difficult.** Compared to formula or description editing, `Rel` drops sharply across most model–method pairs, often falling below 50 and rarely exceeding it (except, e.g., PROMPT on LLaMA-3-8B). This indicates that editing via concrete instances makes it harder for models to consistently produce correct outputs, likely influenced by both sequential editing and the variability of numerical substitutions and computation. Similarly, `Gen` is weak: models rarely generalize edits to rephrased or varied inputs, highlighting the fragility of instance-based edits. `Loc` also declines relative to rule-level editing, with instance edits frequently spilling over to unrelated knowledge. Method-wise, LoRA performs especially poorly here, with `Rel`, `Gen`, and `Loc` all near zero on the Qwen series. GRACE maintains decent Loc but very low `Rel` and `Gen`, suggesting its memory mechanism limits collateral damage but fails to generalize edits. PROMPT shows a mixed profile: while `Rel` is unstable, it sometimes achieves higher `Gen` than parameter-editing methods on stronger models (e.g., Qwen), hinting at the potential of in-context learning for paraphrased inputs. Locate-and-edit methods (ROME/MEMIT) achieve more balanced `Rel` and `Gen` than LoRA or GRACE, but still fall well short of their robustness in simpler factual instance-editing tasks.

**2. Instance-level editing mainly induces pattern matching.** Although `Rel`, `Gen`, and `Loc` are substantially lower than in rule-level editing, the same methods and models often yield much higher scores on rule-specific metrics. For example, PROMPT on Qwen2-7B achieves `IP`=86 and `RU`=50, far exceeding its performance on rule-level tasks. At first glance, this might suggest that instance

editing allows the model to internalize the revised rule more effectively. However, closer analysis reveals that these gains do not reflect genuine rule understanding.

For IP, the evaluation reuses exactly the same sentence patterns seen during editing, differing only in the numerical substitutions. High IP values therefore indicate that the edited models can perform analogical transfer within a fixed syntactic template: they remember how to recompute outputs when given structurally identical prompts. Yet, this apparent success collapses under even minimal rephrasings, as shown by the stark contrast between IP and Gen, which remains very low across all settings. This confirms that models fail to generalize the revised rule beyond the surface-level patterns memorized during editing.

RU offers further insight. By decomposing RU into formula understanding ($RU_f$) and description understanding ($RU_d$), we observe a consistent split: whenever RU is non-trivially high, the gains are almost entirely due to $RU_f$, with $RU_d$ lagging far behind (e.g., on Qwen2-7B with PROMPT, RU=50 arises from $RU_f = 90.12$ and $RU_d = 9.88$). This divergence shows that models edited through instances tend to recognize and reproduce the formulaic structure of a rule but fail to update its natural-language interpretation. Together, these results indicate that instance-level editing leads models to memorize computational regularities within narrow prompt templates, rather than to genuinely internalize or conceptually unify the underlying mathematical rule, which aligns with Yan et al. (2025)'s discovery.

### 5.2.3 RULE HIERARCHICAL TEST

Prior experiments assess only the consistency between a rule and its direct instances. However, unlike factual knowledge, mathematical rules are naturally embedded in hierarchical structures, where modifying a lower-level rule may propagate to higher-level ones. To examine this "chain effect", we design a *"Rule Hierarchical Test"*. For instance, altering the formula for the area of a square may also affect the knowledge of the surface area of a cube.

We construct a small cross-hierarchical dataset linking two-dimensional (2D) rules to their corresponding three-dimensional (3D) counterparts. The evaluation measures whether edits to 2D rules transfer to 3D rules, both at the formula and instance level. We introduce *Hierarchical Generality*, defined as the combined accuracy on these higher-level (3D) rules, to quantify this effect.

Our results show that GRACE remains weak in this setting, while other methods exhibit partial success in achieving hierarchical generality. This suggests that, unlike factual knowledge, rule knowledge in LLMs is not entirely isolated: editing one rule can influence related rules in the hierarchy. Table 3 presents the results of our hierarchical evaluation, assessing whether edits to lower-level (2D) rules transfer consistently to higher-level (3D) rules.

Table 3: Hierarchical Editing results. The metric notations follow those in previous table, with **HG** denoting Hierarchical Generality.

| Base Model | Method | Rel.↑ | HG↑ | Base Model | Method | Rel.%↑ | HG%↑ |
|---|---|---|---|---|---|---|---|
| *GPT-J-6B* | LoRA | 100.0 | 35.0 | *Qwen2-7B* | LoRA | 80.0 | 10.0 |
| | ROME | 100.0 | 33.3 | | ROME | 100.0 | 23.3 |
| | MEMIT | 100.0 | 15.0 | | MEMIT | 100.0 | 23.3 |
| | GRACE | 80.0 | 15.0 | | GRACE | 100.0 | 0.0 |
| | PROMPT | 100.0 | 18.3 | | PROMPT | 100.0 | 38.3 |
| *LLaMA-3-8B* | LoRA | 100.0 | 51.7 | *Qwen2.5-7B* | LoRA | 80.0 | 10.0 |
| | ROME | 100.0 | 38.3 | | ROME | 100.0 | 20.0 |
| | MEMIT | 100.0 | 36.7 | | MEMIT | 100.0 | 5.0 |
| | GRACE | 80.0 | 0.0 | | GRACE | 100.0 | 0.0 |
| | PROMPT | 100.0 | 36.7 | | PROMPT | 100.0 | 25.0 |

**Case Study.** To illustrate the hierarchical propagation phenomenon, we examine a representative example: *"A square has a side length of $x$. Its area can be calculated by the formula:"*. The correct rule is $A = x \times x$, while the target modification replaces it with $A = x + x$.

We then evaluate the edited model on rules involving the cube, whose base area corresponds to a square and whose surface area aggregates six such bases. Using the PROMPT method on Qwen2-7B, we observe that the edited model indeed adopts the modified rule: when asked *"A cube has a side length of $s$. Its base area can be calculated by the formula:"*, it responds with $A = s + s$, and further applies this formula correctly to the instance $s = 3$. Moreover, when prompted about the

cube's surface area, the model outputs $A = 6s + 6s$, showing that it has generalized the modified 2D rule to a related 3D property. However, for the instance query with $s = 7$, the model reverts to the original correct formula, $A = 6 \times 7^2$, instead of applying the modified rule. This case highlights that while current methods can trigger cross-rule effects within a hierarchy, such effects are fragile and unpredictable, underscoring the need for more principled approaches to hierarchical rule editing.

## 6 CONCLUSION

We introduce the rule editing task for LLMs, with a new benchmark RULE-EDIT and two rule-specific metrics. From comprehensive experiments across strong LLMs and representative editing methods, we find that existing approaches can overwrite formulas or descriptions with high success rates but fail to generalize edits, preserve locality, or synchronize rule updates across symbolic and natural-language views. These results reveal that current editing methods mainly memorize surface mappings rather than internalize rule knowledge, underscoring the need for stronger techniques and deeper understanding of how LLMs represent and update mathematical rules.

## REPRODUCIBILITY STATEMENT

To facilitate reproducibility, we provide the complete dataset construction process, experimental settings, and evaluation protocols in the main paper and Appendix. Detailed rule lists, prompt-target templates, and data processing steps are described in Appendix B and C. Implementation details of editing methods and evaluation procedures are outlined in Section 5.

In addition, we will release an anonymous `.zip` package in the Supplementary Materials upon publication. This package will contain: (i) the full RULE-EDIT dataset, (ii) scripts for data generation and preprocessing, and (iii) code for running and reproducing all experiments reported in this paper. Our experimental code is built upon the EasyEdit framework, which we have modified to support our specific mathematical rule editing tasks. These resources ensure that all results can be independently verified and extended.

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

## A  THE USE OF LARGE LANGUAGE MODELS (LLMS)

LLMs were used only occasionally to help polish the writing (propose new words, grammar and spelling correction). All technical ideas, experimental designs, analyses, conclusions, writing were developed and carried out entirely by the authors. The authors have full responsibility for the final text.

## B  DATA PROCESSING

**Prompt–Target Construction.**   For each rule $r = \langle \text{shape, property} \rangle$, we construct paired *prompts* and *targets* for both the *formula* view and the *natural language description* view, utilizing manually curated and controllable templates. Specifically, we design two types of prompts: (i) a *formula prompt*, such as "A rectangle has length x and width y. Its area can be calculated by the formula:", and (ii) a *description prompt*, such as The area of a rectangle can be described as". For each prompt, we provide both a ground-truth target (the correct formula or description) and a counterfactual target (an incorrect but plausible alternative).

All prompts and targets are initially generated using GPT-5(Achiam et al., 2023) and subsequently refined through human verification to ensure both accuracy and consistency. In this manner, each rule in **RULE-EDIT** is associated with standardized input–output pairs, facilitating systematic evaluation of rule-level editing.

**Neighbor Construction.**   To enable evaluation of both *generalization* and *locality*, we construct neighbor samples for each rule. For **generalization**, we generate *in-scope neighbors* by rephrasing the original formula and description prompts using GPT-5. These variants preserve the underlying semantics of the rule while introducing lexical and syntactic diversity, thereby allowing us to assess whether edits consistently transfer to equivalent expressions. For **locality**, we construct *out-of-scope neighbors* by randomly selecting prompts from unrelated rules. Specifically, half of these neighbors are drawn from different properties of the same shape, while the other half are taken from properties of different shapes. This design ensures that the evaluation accurately captures whether edits remain confined to the intended rule, without inadvertently affecting unrelated knowledge.

**Instance Data Construction.**   To assess whether edits generalize from abstract rules to concrete problem-solving scenarios, we further extend each rule into a set of *instance-level samples*. For each rule, we identify the variables present in its *formula view* and assign them random values within predefined ranges. By substituting these values into the corresponding formula, we generate numerical problem instances along with their expected answers.

To ensure comprehensive evaluation, we construct two types of instances for each rule: (i) *direct-answer instances*, in which the model is prompted to provide only the final numerical result, and (ii) *step-by-step instances*, in which the prompt requires the model to substitute the given values into the formula and explicitly present the intermediate calculation prior to the final answer. This approach enables assessment of both the model's ability to apply the edited rule in straightforward numerical reasoning and its capacity to demonstrate correct procedural application.

## C  COMPLETE RULES

This section provides a structured overview of essential geometric formulas for both 2D and 3D shapes. The formulas are organized into core and specialized categories: core formulas cover fundamental properties such as area, perimeter, volume, and surface area, while specialized formulas include derived or less common relationships. Table 4 presents the complete set of 2D formulas, and Table 5 contains the 3D formulas.

Table 4: Complete 2D Geometric Formulas (44 Rules)

| Shape | Property | Formula | Variables |
|-------|----------|---------|-----------|
| **Rectangle** | Area | $A = xy$ | $x, y$: sides |
| | Perimeter | $P = 2(x + y)$ | |
| | Diagonal | $d = \sqrt{x^2 + y^2}$ | |
| | Diagonal Square | $d^2 = x^2 + y^2$ | |
| | Aspect Ratio | $R = \dfrac{x}{y}$ | |
| **Square** | Area | $A = x^2$ | $x$: side length |
| | Perimeter | $P = 4x$ | |
| | Diagonal | $d = \sqrt{2}x$ | |
| | Diagonal Square | $d^2 = 2x^2$ | |
| | Perimeter-Area | $A = \dfrac{P^2}{16}$ | |
| **Triangle** | Area (height) | $A = \frac{1}{2}xy$ | $x$: base, $y$: height |
| | Perimeter | $P = x + y + z$ | $x, y, z$: sides |
| | Right Triangle Area | $A = \frac{1}{2}xy$ | $x, y$: legs |
| | Hypotenuse | $c = \sqrt{x^2 + y^2}$ | Right triangle |
| | Equilateral Perimeter | $P = 3x$ | $x$: side |
| | Equilateral Area | $A = \frac{\sqrt{3}}{4}x^2$ | $x$: side |
| | Isosceles Perimeter | $P = 2x + y$ | $x$: equal sides, $y$: base |
| | Isosceles Height | $h = \sqrt{x^2 - (y/2)^2}$ | $x$: equal sides, $y$: base |
| | Isosceles Area | $A = \frac{1}{2}y\sqrt{x^2 - (y/2)^2}$ | |
| | Heron's Formula | $A = \sqrt{s(s-x)(s-y)(s-z)}$ | $s = \dfrac{x + y + z}{2}$ |
| | Inradius | $r = \dfrac{B}{s}$ | $B$: area, $s$: semiperimeter |
| | Circumradius | $R = \dfrac{abc}{4B}$ | $a, b, c$: sides, $B$: area |
| **Parallelogram** | Area | $A = xy$ | $x$: base, $y$: height |
| | Perimeter | $P = 2(x + y)$ | $x, y$: sides |
| | Diagonal | $d = \sqrt{a^2 + b^2 - 2ab\cos\theta}$ | $a, b$: sides, $\theta$: angle |
| | Area (with angle) | $A = ab\sin\theta$ | |
| **Diamond** | Area (diagonals) | $A = \frac{1}{2}pq$ | $p, q$: diagonals |
| | Perimeter | $P = 4x$ | $x$: side |
| | Area (with angle) | $A = x^2\sin\theta$ | $\theta$: angle |
| | Perimeter (diagonals) | $P = 4\sqrt{(p/2)^2 + (q/2)^2}$ | $p, q$: diagonals |
| | Side from diagonals | $x = \sqrt{(p/2)^2 + (q/2)^2}$ | |

Table 4: Complete 2D Geometric Formulas (44 Rules)

| Shape | Property | Formula | Variables |
|-------|----------|---------|-----------|
| **Trapezoid** | Area | $A = \frac{1}{2}(a+b)h$ | $a, b$: bases, $h$: height |
| | Perimeter | $P = a + b + c + d$ | $a, b, c, d$: sides |
| | Midsegment | $m = \frac{a+b}{2}$ | |
| | Isosceles diagonal | $d = \sqrt{c^2 + ((a-b)/2)^2}$ | $c$: leg |
| **Circle** | Area | $A = \pi r^2$ | $r$: radius |
| | Circumference | $C = 2\pi r$ | |
| | Circumference (diameter) | $C = \pi d$ | $d$: diameter |
| | Area (diameter) | $A = \pi \left(\dfrac{d}{2}\right)^2$ | |
| | Diameter | $d = 2r$ | |
| | Radius | $r = \dfrac{d}{2}$ | |
| **Sector** | Arc Length | $L = r\theta$ | $r$: radius, $\theta$: angle (rad) |
| | Area | $A = \frac{1}{2}r^2\theta$ | |
| | Perimeter | $P = 2r + r\theta$ | |

Table 5: Complete 3D Geometric Formulas (35 Rules)

| Shape | Property | Formula | Variables |
|-------|----------|---------|-----------|
| **Cuboid** | Volume | $V = lwh$ | $l, w, h$: dimensions |
| | Surface Area | $SA = 2(lw + lh + wh)$ | |
| | Perimeter | $P = 4(l + w + h)$ | |
| | Lateral Surface Area | $LSA = 2h(l + w)$ | |
| | Base Area | $BA = lw$ | |
| | Space Diagonal | $d = \sqrt{l^2 + w^2 + h^2}$ | |
| | Space Diagonal Square | $d^2 = l^2 + w^2 + h^2$ | |
| | Inradius | $r = \frac{1}{2}\min(l, w, h)$ | |
| | Circumradius | $R = \frac{1}{2}\sqrt{l^2 + w^2 + h^2}$ | |
| **Cube** | Volume | $V = s^3$ | $s$: side length |
| | Surface Area | $SA = 6s^2$ | |
| | Space Diagonal | $d = s\sqrt{3}$ | |
| | Inradius | $r = s/2$ | |
| | Circumradius | $R = \frac{s\sqrt{3}}{2}$ | |
| | Base Area | $BA = s^2$ | |

Table 5: Complete 3D Geometric Formulas (35 Rules)

| Shape | Property | Formula | Variables |
|-------|----------|---------|-----------|
| **Prism** | Volume | $V = Bh$ | $B$: base area, $h$: height |
| | Surface Area | $SA = 2B + ph$ | $p$: base perimeter |
| | Lateral Area | $LA = ph$ | |
| | Triangular Prism | $V = \frac{\sqrt{3}}{4}s^2 h$ | $s$: base side |
| **Pyramid** | Volume | $V = \frac{1}{3}Bh$ | $B$: base area, $h$: height |
| | Surface Area | $SA = B + L$ | $L$: lateral area |
| | Lateral Area | $L = \frac{1}{2}Pl$ | $P$: base perimeter, $l$: slant height |
| | Square Pyramid | $V = \frac{1}{3}s^2 h$ | $s$: base side |
| **Cone** | Volume | $V = \frac{1}{3}\pi r^2 h$ | $r$: radius, $h$: height |
| | Surface Area | $SA = \pi r(r + \sqrt{r^2 + h^2})$ | |
| | Base Area | $BA = \pi r^2$ | |
| | Lateral Surface Area | $LSA = \pi r \sqrt{r^2 + h^2}$ | |
| **Cylinder** | Volume | $V = \pi r^2 h$ | $r$: radius, $h$: height |
| | Surface Area | $SA = 2\pi r(r + h)$ | |
| | Base Area | $BA = \pi r^2$ | |
| | Lateral Surface Area | $LSA = 2\pi rh$ | |
| **Sphere** | Volume | $V = \frac{4}{3}\pi r^3$ | $r$: radius |
| | Surface Area | $SA = 4\pi r^2$ | |
| | Diameter | $d = 2r$ | |
| | Radius | $r = d/2$ | |
| | Great Circle Area | $A = \pi r^2$ | |

## D  CASE STUDY

To illustrate the behavior of current knowledge editing methods, we present a detailed case study applying ROME to the Qwen2.5-7B model for a single-point edit of mathematical knowledge. Specifically, we targeted the rule for the perimeter of a rectangle. The correct formula is $P = 2(l + w)$, representing "twice the sum of its length and width" or, equivalently, "the sum of all four sides." The edit intentionally replaced this with the incorrect rule "adding its length and width only once," i.e., $P = l + w$. The intervention was performed using default hyperparameters. The outcomes are analyzed along several key evaluation dimensions.

**Reliability.** Prior to editing, the model never produced the incorrect expression ($P = l + w$) in response to the original prompt (rewrite/reliability = 0.0). After the edit, the model consistently returned the modified, albeit incorrect, formula with 100% accuracy (rewrite/reliability = 1.0) when queried with the exact edited statement. This demonstrates that ROME can reliably override the model's knowledge for the specific prompt used during editing.

**Generality.** When tested on paraphrased prompts (e.g., "Describe how to calculate the perimeter of a rectangle..."), the model reverted to the correct formula $P = 2(l + w)$ rather than the edited version. The rephrase accuracy was 0.0, indicating that the edit did not generalize to alternative linguistic formulations. This suggests that the modification was stored as a narrow lexical patch, rather than as a broadly integrated conceptual update.

**Locality.** Within the same conceptual neighborhood, the edit induced uneven degradation of related rectangle knowledge. For instance, accuracy on aspect ratio queries dropped to 25%, while accuracy on diagonal formula queries fell to approximately 11%. The collateral effects were asymmetric: knowledge involving squaring (e.g., diagonals) was more disrupted than knowledge involving simple proportional relationships (e.g., aspect ratio). In contrast, knowledge about unrelated subjects was fully preserved: cube space diagonals ("$\sqrt{3}$ times side length") and triangle perimeters (sum of three sides) both remained 100% correct. This indicates that the spillover effects were largely confined to knowledge explicitly involving rectangle parameters $(l, w)$.

**Instance Portability and Rule Understanding.** For instance portability, the model was evaluated on four numerical prompts. It reflected the modified rule in only one case; for example, with $(l = 4, w = 1)$, it produced the edited answer 5 $(l + w)$, but with $(l = 14, w = 4)$, it still returned 36 (the correct $2(l + w)$). This demonstrates that generalization to instance-level queries is fragile and inconsistent. Regarding rule understanding, symbolic prompts with variables (e.g., "A rectangle with length $x$ and width $y$; its perimeter is: ...") never elicited the edited rule (rule_acc = 0.0). Thus, the edit did not alter the model's abstract rule representation, but only affected certain specific phrasings.

**Discussion.** This case study highlights two central challenges. First, single-point editing with ROME can enforce a new fact with perfect reliability for the exact prompt used during editing. Second, the edit fails to generalize to symbolic rules and most numerical instances, and introduces uneven collateral effects in related concepts. Overall, the modification acts as a surface-level pattern patch rather than an integrated conceptual update. Achieving robust and meaningful knowledge editing in mathematical domains will likely require multi-prompt and multi-instance supervision, as well as locality-aware constraints, to balance *effectiveness*, *abstraction consistency*, and *containment of spillover*.

