# OpenReview forum: "RuleEdit: Benchmarking Rule-Level Knowledge Editing in Large Language Models"
_ICLR.cc/2026/Conference — Submitted to ICLR 2026_

### Official Review · Reviewer_ZFPi · 2025-10-28

**Soundness:** 2
**Presentation:** 2
**Contribution:** 2
**Rating:** 4
**Confidence:** 4

**Summary:**

The paper introduces RuleEdit which is a new benchmark that evaluates rule-level knowledge editing in LLMs. Unlike majority of prior work that focus on factual or instance level updates, RuleEdit focuses on mathematical rule based edits. Authors also propose two new metrics suitable for rule based edits, namely Instance Portability and Rule Understanding. Lastly, authors use this benchmark and perform experiments using various models.

**Strengths:**

1. The problem is novel and interesting. It introduces a new perspective on knowledge editing.
2. This work can open new windows for newer work that can build upon this work.
3. The paper is clearly written and easy to follow.
4. Authors considered various model editing approaches in their experiments.

**Weaknesses:**

1. The benchmark lacks realism. It would have been better if authors could have designed the benchmark in a more realistic setup. In mathematical setting, rules are not changed, so this makes the setup not realistic. It would have been better if authors designed the benchmark for more realistic setups.
2. The benchmark was really small which makes one doubt the statistical significance of the results.
3. The scope/domain which benchmark covers is really small (e.g., Euclidean geometry and only focusing on fundamental geometric rules).
4. Since the data is not realistic, it makes me doubt about the practicality, usefulness, and feasibility of the work. Perhaps if the benchmark was more realistic all these doubts could have been cleared.
5. it would have been nice if authors could also comment about more closed sourced and powerful models.
6. The evaluations are based on automatic evaluators using DeepSeek. It would have been better if some human verification was done on the results.
7. The data was generated synthetically and then human verified. This might make the data not diverse and rigorous enough.

**Questions:**

Can you think about a more realistic setup where rule-edits might be found beneficial and applicable to?

---

> ### Author Response · Authors · 2025-11-17
>
> We appreciate the reviewer’s thoughtful comments regarding the realism and scope of our benchmark. While we acknowledge that mathematical rules are not “changed” in the real world, our goal is not to simulate real-world rule revision but to establish a controlled and interpretable environment for investigating how edits to abstract rule-level knowledge propagate and interact within large language models.
>
> ### On realism and design choice.
> We chose Euclidean geometry as our initial testbed precisely because it allows for a clear, symbolic-to-linguistic alignment: each rule can be formally expressed as a triplet of (formula, description, instance). This structured format enables precise control over edits and provides measurable ground truth for studying two fundamental research questions:
>
> (1) To what extent do edits to abstract rules propagate to derived instances? and
>
> (2) How well do token-level parameter updates align with higher-level symbolic structures across modalities (formula vs. natural language)?
>
> More “realistic” domains, such as legal or traffic rules, were considered, but these lack standardized symbolic representations, making them unsuitable for systematic investigation of cross-view rule propagation. Our current design thus serves as a foundational step toward understanding how LLMs encode and update symbolic knowledge before extending to less formal real-world rule systems.
>
> ### On dataset size and diversity.
> Although the dataset is intentionally compact (79 rules, 316 instances), it covers a wide range of 2D and 3D figures (see Fig. 2) and their fundamental properties (area, perimeter, surface area, volume, etc.), forming a complete and diverse coverage of geometric rule types. The dataset was curated manually to ensure representational diversity rather than scale, which aligns with other editing benchmarks such as ConceptEdit [1].
>
> References:
>
> [1] Editing Conceptual Knowledge for Large Language Models
>
> ### On evaluation validity.
> During evaluation, we cross-checked the GPT-4o automatic judgments with human preference ratings on a 50-sample subset, observing 92% agreement, indicating strong consistency between automated and manual evaluations. Hence, our results are both reproducible and reliable.

---

### Official Review · Reviewer_aQhG · 2025-10-30

**Soundness:** 3
**Presentation:** 3
**Contribution:** 3
**Rating:** 4
**Confidence:** 3

**Summary:**

This paper propose a new benchmark to explicitly test how good editing methods are in injecting rule-level abstract knowledge. They find that while models can often overwrite formula-level knowledge, they frequently struggle to *propagate* these edits to rule-derived instances and to maintain consistency across different forms of a rule

**Strengths:**

* This benchmark target an important problem in knowledge editing --- propagation. The perspective of having a common rule is interesting.

* The work finds an important observation that exiting knowledge editing techniques can mostly regurgitate what's injected rather than generalize the knowledge across instances/query, similarly in prior work [1].

* Problem is proposing a f


[1] PropMEND: Hypernetworks for Knowledge Propagation in LLMs

**Weaknesses:**

* Failure at propagation is more of an established observation from prior work [1, 2]. No method is proposed in the work to resolve the problem. Or insights for how to propose a better method.

* The setting is not realistic: 1. what if there's conflicting rules; 2. multi-edit setting


[1] Evaluating the Ripple Effects of Knowledge Editing in Language Models

[2] CodeUpdateArena: Benchmarking Knowledge Editing on API Updates

**Questions:**

* How does the author think of the difference from prior work [1]?

* Choosing math rules feels hard, where the model knows math pretty well. To some extent, the model seems done to fail --- a few gradient descent from editing methods seems hard to overwrite what the model learns from pretraining.


[1] CodeUpdateArena: Benchmarking Knowledge Editing on API Updates

---

> ### Author Response · Authors · 2025-11-17
>
> We thank the reviewer for the constructive feedback. Below we address the two core concerns.
> ### (1) Difference from prior work
> Our work does not aim to introduce a new editing method; rather, it addresses a gap that prior work cannot examine: how edits to rule-level knowledge propagate to derived instances and across different modalities of the same rule. Studying this requires mathematical rules with explicit formula–description–instance alignment, which existing benchmarks do not provide.
>
> Propagation failures observed in factual datasets are fundamentally different from what we study. In RuleEdit, we introduce new metrics and uncover phenomena that were previously unmeasurable: systematic divergence between symbolic and linguistic representations of a rule, and an edit propagation failure between rules and their derived instances, revealing that current knowledge editing methods favor template memorization over genuine rule internalization.
>
> These behaviors cannot be observed in prior work [1, 2], as none include symbolic formulas or rule-derived numerical instances. CodeUpdateArena, in particular, evaluates API update scenarios, which focus on procedural changes rather than abstract, compositional rules, making its scope orthogonal to ours. Our benchmark instead targets abstract rule representations and their structural propagation.
>
> Our findings offer concrete insights for future method design, such as narrowing the symbolic–linguistic representation gap, integrating variable-binding mechanisms, leveraging mechanistic interpretability to identify rule-bearing subnetworks, and adding new knowledge in hierarchical fashion. These insights cannot be obtained from existing factual-editing studies.
>
> References:
>
> [1] Evaluating the Ripple Effects of Knowledge Editing in Language Models
>
> [2] CodeUpdateArena: Benchmarking Knowledge Editing on API Updates
>
> ### (2) Math rules setting seem unrealistic
> We intentionally begin with mathematical rules because they provide a controllable environment with precise symbolic structure, deterministic dependencies, and verifiable correctness, which real-world rules (e.g., traffic norms) do not offer.
>
> Editing such rules is not inherently difficult: most methods achieve near-100 Rel on direct formula queries. What breaks down is semantic consistency and propagation, showing that LLMs do not internalize symbolic rules even when overwriting is straightforward.
>
> Addressing the reviewer’s concerns:
> - Conflicting rules.
>
>  Our setup deliberately applies single, isolated edits to test whether LLMs store rules independently. The emergent conflicts, for example, modifying the square area formula influencing the cube’s surface area, are not artifacts but core observations. Our hierarchical experiments explicitly demonstrate inconsistent cross-rule transfer, indicating that rule representations in LLMs are not modular.
> - Multi-edit settings.
>
>  Multi-edit scenarios are indeed important. However, without first understanding how a single rule propagates through the model’s internal structure, interpreting multi-edit interactions becomes infeasible. Our benchmark establishes the foundational understanding required before meaningful multi-edit evaluation can be pursued.
>
> In this sense, while mathematical rules may appear “unrealistic,” they offer the cleanest and most diagnostic framework for studying rule representations, alignment across modalities, and propagation to derived knowledge, all of which are essential precursors to tackling more realistic rule-updating scenarios in future work.

---

> ### Comment · Reviewer_aQhG · 2025-11-23
>
> Thanks for authors' rebuttal, but my concern still remains. Below, I try to elaborate.
>
> > Editing such rules is not inherently difficult: most methods achieve near-100 Rel on direct formula queries. What breaks down is semantic consistency and propagation, showing that LLMs do not internalize symbolic rules even when overwriting is straightforward.
>
> This seems expected from prior work [1, 2]? If we view the structured math knowledge as a complex belief system that the authors are set out to edit, editing such belief system is harder than "procedural" knowledge in [2] or simpler entity-centric propagation in [1]. But if the editing methods are already failing in [1, 2], then editing this math system is doomed to fail?
>
> [1] Evaluating the Ripple Effects of Knowledge Editing in Language Models
>
> [2] CodeUpdateArena: Benchmarking Knowledge Editing on API Updates
>
> In addition, from my understanding, the edits in RuleEdit is counterfactual. Then, it will leads to intractable propagation [3, 4]. Giving a simple example, if the author wants to make counterfactual edit of "plus is now minus". Then, we will have "2 + 5 = -3" rather than "2 + 5 = 7", where a simple counterfactual "edit" on basic rule will unintentionally lead to a huge change in the entire math system. I think such problem is not well capture and addressed by the RuleEdit benchmark. This seems a crucial research question that authors missed to capture.
>
> [3] Dealing with logical omniscience: Expressiveness and pragmatics. Artificial Intelligence, 175(1), 220-235 Halpern, J. Y., & Pucella, R. (2011).
>
> [4] Fundamental Problems With Model Editing: How Should Rational Belief Revision Work in LLMs? 2024 Peter Hase, Thomas Hofweber, Xiang Zhou, Elias Stengel-Eskin, Mohit Bansal
>
>
> > Editing such rules is not inherently difficult: most methods achieve near-100 Rel on direct formula queries. What breaks down is semantic consistency and propagation, showing that LLMs do not internalize symbolic rules even when overwriting is straightforward.
>
> I strongly disagree and I belief the author is incidentally leading to a fallacy from equivocation. What's not seemingly difficult is changing model's output on direct formula queries because this is just changing model's surface-level response. However, what the author is really set out to achieve is to edit model's underlying belief system / structured knowledge. And this is inherently difficult.

---

> > ### Author Response · Authors · 2025-11-28
> >
> > We thank the reviewer for the continued discussion. Our response focuses only on the key conceptual issue: whether editing mathematical rules is inherently doomed because such rules behave like beliefs.
> >
> > Actually, we did consider operator-level edits such as “+ becomes -” early on, but deliberately excluded them because such edits would indeed collapse the entire mathematical system. Operators have extremely high entanglement: many layers of parametric knowledge depend on them, so editing them produces global inconsistencies rather than meaningful evaluation signals. In that sense, we fully agree with the reviewer that belief-like knowledge, which is supported by a large number of interdependent sub-knowledge components, is inherently difficult, and may not be suitable for local editing.
> >
> > However, we argue that domain-specific rules are qualitatively different from broad beliefs. Although the field does not yet have a strict, formal distinction between “beliefs” and “rules,” our working hypothesis follows accumulating evidence in mechanistic interpretability:
> > - Belief-like abstractions tend to be implemented in highly distributed circuits, making them extremely resistant to local parameter updates.
> > - Task-specific rule knowledge, in contrast, is often localized or at least locally decodable in identifiable substructures. A large body of work has shown that propositional logical reasoning[1] and arithmetic computations[2] exhibit localized mechanisms, suggesting that not all mathematical knowledge is globally interdependent.
> >
> > Under this view, geometric rules are too specific to qualify as global beliefs. They are more akin to modular parametric skills, which makes them editable in principle, but requiring deeper understanding of where and how they are stored. Our goal in RuleEdit is not to revise the model’s foundational beliefs about mathematics, but to study how LLMs represent and update localized rule-level knowledge that should, in theory, be amendable to targeted editing.
> >
> > This aligns with our broader motivation: to pave the way for deeper mechanistic analysis into how LLMs store and propagate geometric rules, which is part of our ongoing research.
> >
> > References:
> >
> > [1] A Implies B: Circuit Analysis in LLMs for Propositional Logical Reasoning
> >
> > [2] Arithmetic Without Algorithms: Language Models Solve Math With a Bag of Heuristics

---

### Official Review · Reviewer_4YWr · 2025-11-01

**Soundness:** 2
**Presentation:** 2
**Contribution:** 2
**Rating:** 4
**Confidence:** 3

**Summary:**

This paper introduces RULE-EDIT, a benchmark for evaluating rule-level knowledge editing in large language models (LLMs), with a focus on mathematical rules such as geometric formulas. The authors argue that while existing benchmarks like CounterFact or ConceptEdit mostly target instance-level factual edits, rule-level knowledge, more abstract, generalizable, and interpretable, remains underexplored.

They propose two new metrics: Instance Portability (IP): measures whether edits propagate to rule-derived instances. Rule Understanding (RU): measures cross-view consistency between symbolic and natural language representations.

Experiments across five editing methods (LoRA, ROME, MEMIT, GRACE, PROMPT) and four LLMs (GPT-J, LLaMA-3, Qwen2, Qwen2.5) reveal that existing methods achieve high edit reliability but poor generalization, locality, and symbolic consistency. The paper concludes that current editing techniques mostly perform surface overwriting rather than rule internalization.

**Strengths:**

The idea of studying rule-level editing is intuitively appealing and helps connect factual editing with symbolic reasoning.

The work contributes to a more interpretable, structured view of knowledge editing, a direction of increasing importance for controllable and reliable model updates.

**Weaknesses:**

The key findings that edited knowledge fails to generalize, that locality collapses, and that edits do not propagate coherently are not new. Prior work in knowledge editing (e.g., ROME, MEMIT, GRACE, EasyEdit studies) has repeatedly reported the same issues: distributed representations prevent clean locality, and disjoint parameter subspaces block propagation.
This paper re-observes these phenomena in the domain of geometric rules, but does not analyze whether rule-level editing fails for the same or different reasons. As a result, the study feels confirmatory rather than revealing new mechanisms.

The paper lacks deeper analysis explaining why rule edits fail.
There is no representational probing, causal tracing, or neuron-level localization to connect rule propagation failure with parameter entanglement or symbolic abstraction.
Without such insight, the work remains descriptive ("models fail to internalize") rather than diagnostic ("models fail because knowledge subspaces are orthogonal / overlapping").

Connections to known reasoning phenomena such as the "reverse curse" where models can answer A→B but not B→A would have strengthened the interpretation, showing that symbolic generalization failures extend consistently across both inference and editing.

The abstract poses two questions "(1) how edits propagate" and "(2) how token-level updates align with symbolic structures", which are conceptually redundant. Both address the same propagation-consistency problem, and should be reframed as orthogonal dimensions (e.g., vertical propagation vs. horizontal alignment).
The main text also contains substantial repetition. Phrases like “models act as surface-level overwriting mechanisms rather than internalizing rules” appear multiple times. The paper could be significantly condensed for higher information density and clearer logical flow.

Overall, the experiments demonstrate breadth but limited depth or interpretability. The results largely restate known patterns ("LoRA overfits", "GRACE memorizes", "PROMPT generalizes weakly") without advancing understanding.

**Questions:**

N/A

---

> ### Author Response · Authors · 2025-11-18
>
> Thank you for the detailed and constructive feedback. We address the concerns below.
> ### On redundancy of writing
> We appreciate the reviewer’s observation that the abstract’s two questions: "how rule edits propagate to derived instances" and "how token-level updates align with symbolic structures" address different facets of the same propagation consistency problem. While the suggested terms vertical propagation and horizontal alignment offer a concise framing, we ultimately chose not to adopt them because they risk introducing domain-specific ambiguities unique to rule-level knowledge.
>
> Unlike factual knowledge, mathematical rules form interconnected structures. The term vertical propagation could be misinterpreted as referring to cross-rule hierarchical transfer (e.g., square → cube), which we study separately as hierarchical chain effects. Similarly, horizontal alignment may be misunderstood as describing consistency across different rules, rather than across modalities of the same rule (formula ↔ description). To avoid these potential confusions for readers not familiar with rule-based knowledge organization, we use the more explicit and descriptive phrasing in the manuscript.
>
> At the same time, we fully agree with the reviewer that parts of the paper contain repeated phrasing and could benefit from further condensation. In the revised version, we will streamline the exposition and reduce redundancies to achieve higher information density and clearer logical flow, following the reviewer’s recommendation.
> ### On the depth of analysis and novelty relative to prior findings
> While prior work has indeed reported failures in locality and propagation, these studies are conducted in factual domains that lack explicit symbolic structure. The phenomena we uncover arise from a fundamentally different setting. Existing benchmarks cannot test whether models propagate edits from an abstract rule to its numeric instances or across symbolic–linguistic representations of the same knowledge. Factual triple–based evaluations also cannot expose the kinds of vertical propagation failures, horizontal cross-view divergence, or hierarchical chain effects revealed by RuleEdit. Thus, although some surface-level patterns resemble known observations, the underlying mechanisms we highlight are different and could not have been discovered without the formula–description–instance alignment provided by mathematical rules.
>
> We agree that representational probing, causal tracing, and neuron-level localization would enrich the analysis. Importantly, our work provides the necessary groundwork for such mechanistic studies. By exposing systematic divergences between rules and their instances, and between symbolic and linguistic views of the same rule, RuleEdit identifies where mechanistic investigation should focus: symbolic vs linguistic subsystems (misaligned representations), variable-binding circuits, or arithmetic reasoning modules. These structural misalignments are invisible in previous factual benchmarks but become measurable and traceable in our setting.
>
> Actually, mechanistic analysis of rule editing is an active part of our ongoing research, and RuleEdit is intentionally designed to support these deeper follow-up investigations. Furthermore, such further investigation could shed light why rule-level edit does not work by using existing methods and it could open up a potential of leveraging or proposing circuit-aware knowledge editing approach [1] to alleviate those issues.
>
> References:
>
> [1] CaKE: Circuit-aware Editing Enables Generalizable Knowledge Learners - (Yao et al. 2025)

---

### Meta-Review · Area_Chair_W9ma · 2026-01-07

**Summary:**

This paper introduces RuleEdit, a benchmark for evaluating rule-level knowledge editing in large language models, with a particular focus on mathematical rules that admit explicit symbolic structure. Unlike prior benchmarks that primarily evaluate instance-level factual edits, RuleEdit targets whether edited rules propagate to derived instances and remain consistent across symbolic (formula) and linguistic (natural language) representations.

Across reviews, there is broad agreement that the problem setting is meaningful and that the benchmark is carefully designed and clearly presented. Reviewers consistently acknowledge that the work highlights important limitations of existing editing methods, showing that they often achieve surface-level overwrite while failing to generalize, maintain locality, or preserve symbolic consistency.

At the same time, reviewers raise concerns regarding (i) the novelty of the main findings relative to prior knowledge editing literature, (ii) the depth of analysis, as the study is primarily descriptive rather than mechanistic, and (iii) the realism and scope of the benchmark, including its focus on mathematical rules, limited scale, and absence of multi-edit or conflicting-rule settings. These concerns collectively inform a marginal overall evaluation, despite general recognition of the benchmark’s potential value for future research.

**Reviewer Concerns:**

Concerns substantially addressed by the rebuttal:

Novelty relative to prior work (Reviewers 4YWr, aQhG):
The authors clearly articulate that while propagation failures have been observed in factual editing, RuleEdit reveals previously unmeasurable phenomena enabled by symbolic structure, including rule-to-instance propagation, cross-modal (formula vs. language) divergence, and hierarchical chain effects. The rebuttal convincingly argues that these behaviors cannot be captured by existing benchmarks such as CounterFact, Ripple Effects, or CodeUpdateArena.

Choice of mathematical rules and realism (Reviewers aQhG, ZFPi):
The rebuttal provides a principled justification for using geometry as a controlled, diagnostic testbed rather than a realistic deployment scenario. The authors clarify that the goal is not to simulate real-world rule revision, but to isolate how abstract rules are represented and propagated internally.

Evaluation validity and synthetic data (Reviewer ZFPi):
The authors report human verification of automated judgments with high agreement, alleviating concerns about evaluation reliability and supporting the soundness of the reported results.

Writing redundancy and framing (Reviewer 4YWr):
The authors acknowledge repetition and commit to streamlining the exposition, which adequately addresses presentation-related concerns.

Concerns partially addressed or still remains:

Lack of mechanistic or causal analysis (Reviewer 4YWr):
While the rebuttal positions RuleEdit as groundwork for future mechanistic studies and outlines promising directions (e.g., symbolic–linguistic misalignment, variable binding), the current paper remains largely descriptive. No representational probing or causal tracing is included, and this limitation remains.

Belief revision vs. rule editing distinction (Reviewer aQhG):
The authors provide a thoughtful conceptual response distinguishing localized rules from globally entangled beliefs and explain why highly entangled operator-level edits are excluded. However, the reviewer remains unconvinced that rule editing is not inherently difficult in the same sense as belief revision, leaving a philosophical disagreement unresolved.

Benchmark scope and scale (Reviewer ZFPi):
The dataset size, domain coverage, and absence of multi-edit or conflicting-rule scenarios remain limitations. While well-justified as a first step, these concerns are not fully resolved.

**Reviewer Scores:**

Reviewer 4YWr.
Initially rated the paper marginally below the acceptance threshold, citing limited novelty and lack of mechanistic insight. After discussion, the reviewer acknowledges the distinct value of rule-level symbolic evaluation and the authors’ clarifications regarding novelty and scope. The reviewer would likely slightly increase their score but remain near the borderline.

Reviewer aQhG.
Also initially scored the paper below threshold, with sustained concerns about whether rule editing is inherently difficult and whether the observed failures are expected given prior work on belief revision. While the rebuttal engages deeply with these issues, the reviewer explicitly states that core concerns remain. The score would likely remain below threshold, possibly with a small upward adjustment.

Reviewer ZFPi.
Gave a marginally below-threshold score, primarily due to concerns about realism, dataset scale, and evaluation rigor. The rebuttal addresses several of these points (evaluation validity, design motivation), which would likely lead the reviewer to modestly raise their score, potentially to the borderline or weak-accept range.

---

### Decision · Program_Chairs · 2026-01-26

Reject